# Exploring the Association between Oxygen Concentration and Life Expectancy in China: A Quantitative Analysis

**DOI:** 10.3390/ijerph20021125

**Published:** 2023-01-08

**Authors:** Qing Zou, Yingsi Lai, Zhao-Rong Lun

**Affiliations:** 1Department of Medical Statistics, School of Public Health, Sun Yat-Sen University, Guangzhou 510080, China; 2Sun Yat-Sen Global Health Institute, Sun Yat-Sen University, Guangzhou 510080, China; 3State Key Laboratory of Biocontrol, School of Life Sciences, Sun Yat-Sen University, Guangzhou 510275, China

**Keywords:** life expectancy, oxygen concentration, association, hypoxia, aging

## Abstract

The aim of this study was to investigate and quantify the association between oxygen concentration and life expectancy. The data from 34 provinces and 39 municipalities were included in all analyses. Bayesian regression modeling with spatial-specific random effects was used to quantify the impact of oxygen concentration (measured as partial pressure of oxygen) on life expectancy, adjusting for other potential confounding factors. We used hierarchical cluster analysis to group the provinces according to disease burden and analyzed the oxygen levels and the characteristics of causes of death between the clusters. The Bayesian regression analysis showed that the life expectancy at the provincial level increased by 0.15 (95% CI: 0.10–0.19) years, while at the municipal level, it increased by 0.17 (95% CI: 0.12–0.22) years, with each additional unit (mmHg) of oxygen concentration, after controlling for potential confounding factors. Three clusters were identified in the hierarchical cluster analysis, which were characterized by different oxygen concentrations, and the years of life lost from causes potentially related to hypoxia were statistically significantly different between the clusters. A positive correlation was found between oxygen concentration and life expectancy in China. The differences in causes of death and oxygen levels in the provincial clusters suggested that oxygen concentration may be an important factor in life expectancy when mediated by diseases that are potentially related to hypoxia.

## 1. Introduction

Exploring the factors influencing aging is critical for identifying possible targets of intervention for extending the human life span [1]. The biochemical processes that accompany aging are inherently complex [2,3], and a series of studies have suggested that oxygen seems to be inextricably linked to multiple processes of aging [4,5,6]. For example, Synder et al. (2021) found that hypoxia-related mechanisms may contribute to cognitive impairment and interact with aging [7]. Rudloff and his colleagues (2022) showed that hypoxia-induced intrauterine growth restriction increases the risk for cardiovascular, renal, and other chronic diseases in adults [8]. At the molecular level, it is well recognized that chromosome telomeres progressively shorten with increased age [9,10,11], while activating or upregulating telomerases can slow down the speed of this telomere shortening [12,13]. Interestingly, researchers have found that hyperbaric oxygen therapy could significantly increase telomere length and clear senescent cells in aging populations [14], suggesting that oxygen may have an influence on human aging.

Slowing down the rate of aging in populations can increase the overall life expectancy [15], and this is one of the most widely used summary indicators for the overall health of a population [16]. Understand the effect of oxygen concentration on life expectancy can provide important clue regarding the impact of oxygen on population aging. In a very recent study, Lu et al. (2020) found that altitude had a negative effect on life expectancy in China [17]. In contrast, an earlier study by Ezzati et al. (2012) in the U.S. found that living at a higher altitude appeared to have no net effect on life expectancy [18], after adjusting for the influence of other factors (i.e., socio-demographic factors, migration, average annual solar radiation, and cumulative exposure to smoking). The study by Ezzati et al. grouped counties into elevation bands, instead of using actual elevation data in the analysis, which could have led to a loss of information granularity and lower statistical power. In addition, even though areas at a higher altitude tend to have a lower oxygen concentration, there are other factors which could affect the precise oxygen levels at any given altitude. Therefore, it might not be possible to directly relate oxygen levels and life expectancy without a direct determination of oxygen concentration.

To our knowledge, there are few studies that investigate quantitively the association between environmental oxygen concentration and life expectancy. The study by Vold and his colleagues (2015) provided information on the relationship between low oxygen saturation and increased mortality, but it did not address the effects of environmental hypoxia [19]. To better understand the effect of oxygen on life expectancy, China has a large population size which is distributed over a wide range of altitudes from a few meters above sea level to over 4700 m and, therefore, provides a unique living environmental “laboratory” to address this question. For this reason, the aims of this study are to quantify the impact of oxygen concentration on life expectancy using a spatial statistical methodology and to further explore the association between hypoxia and causes of death.

## 2. Materials and Methods

### 2.1. Main Variable

#### 2.1.1. Life Expectancy

The provincial data on life expectancy in 2015 were mainly obtained from the estimation results in Zhou et al. [20]. This study was based on the data of Global Burden of Disease [21] and used the life table method to estimate the life expectancy of 33 province-level administrative divisions (not including Taiwan) in China. Taiwan’s life expectancy data in 2015 were obtained from a news report released by the Global Times [22]. The municipal data on life expectancy were obtained from the official websites of the municipal health commissions, the health statistics bureaus, and the Centers for Disease Control and Prevention (CDCs) during the years from 2012 to 2018. The details of these data sources are listed in Appendix A.

#### 2.1.2. Oxygen Concentration

We collected the data on pressure from the National Meteorological Data Center [23], and calculated oxygen concentration by “Oxygen concentration(mmHg) = pressure (kPa) × 20.93% [24] × 7.5 mmHg/kPa”.

#### 2.1.3. Age-Standardized Years of Life Lost (YLLs) Per 100,000 Population for the Top 20 Level 3 Causes in China

The data on age-standardized YLLs per 100,000 population for the top 20 level 3 causes in China were obtained from the results in Zhou et al. [21]. YLL is one way to measure the mortality impact, which gives higher weight to deaths at younger ages (premature mortality).

### 2.2. Other Variables

We collected data on sunshine, wind speed, relative humidity, and temperature to adjust for the effects of meteorological factors, and calculated (or collected) the GDP per capita, the number of health technicians per 1000 people, and average years of education to measure the effects of economic status, health resources, and educational situation, respectively.

#### 2.2.1. Meteorological Data

There were 839 meteorological surveillance stations that were examined in this study that provided daily observational data. Excluding uninhabited stations, the data from 799 stations were retained. We downloaded the shapefiles (i.e., ADM1 and ADM2) for China from a Database of Global Administrative Areas [25], linking daily observational data for 31 provincial-level administrative divisions and 39 municipalities to the divisions (i.e., ADM1 and ADM2), and averaging them within the corresponding divisions for subsequent statistical analysis. We also averaged the daily observations for Macau, Hong Kong, and Taiwan for subsequent statistical analysis.

#### 2.2.2. GDP Per Capita (Economic Status)

The data on GDP per capita for the 34 province-level administrative divisions in 2015 were collected from the China Statistical Yearbook 2016 or official bureau of statistics. The GDP per capita data for the 39 municipalities were collected from the “2015 National Economic and Social Development Statistical Bulletin” for each region. Detailed sources of the GDP per capita are shown in Appendix A.

#### 2.2.3. Number of Health Technicians Per 1000 People (Health Resources)

The data on health technical personnel per 1000 people for the 34 province-level administrative divisions and 39 municipalities were calculated using the formula “Number of health technicians per 1000 people = Health technical personnel/Number of permanent residents at the end of the year × 1000”. The sources of health technical personnel and the number of permanent residents at the end of the year are shown in Appendix A.

#### 2.2.4. Average Years of Education (Educational Situation)

The data on average years of education for the 34 province-level administrative divisions and 39 municipalities were calculated using the formula “Average years of education = (Number of students in university (college and above) × 16 + Number of students in high school × 12 + Number of students in middle school × 9 + Number of students in primary school × 6 + Populations not attending school × 0)/Number of populations aged six years and over” [26]. This indicator might be a little overestimated, as we used the total years of education at each stage and multiplied it by the number of students to calculate the average years of education, but some students might actually drop out before finishing the education of the corresponding stages. However, the dropout rates in China are low [27], so this impact could be ignored. The sources of data used in this formula are shown in Appendix A.

### 2.3. Statistical Analyses

Firstly, we used the mean ± standard deviation to summarize continuous data following a normal distribution; otherwise, the median (lower quartile–upper quartile) was used. We mapped the geographical distribution of life expectancy and oxygen concentration and created corresponding scatter plots. High altitude was defined as terrestrial elevations over 1500 m [28,29,30], and the body could be in an anoxic state, corresponding to an oxygen concentration around 140 mmHg. Therefore, we divided all the data into two groups with the cut-off value (oxygen concentration = 140 mmHg). The Mann–Whitney U test was used to compare the differences between two groups.

#### Statistical Model

Prior to modeling, Pearson correlation analyses were carried out and univariate regression models were developed. If a pair of variables had a correlation coefficient > 0.7, the variable with the highest value of the deviance information criteria in the univariate model was excluded. Furthermore, the remaining variables were selected using the stepwise method.

We adopted Bayesian regression models with spatial-specific random effects to quantify the impact of oxygen concentration on life expectancy, adjusting for the effects of other factors (e.g., economic status, medical resources, and educational situation). In this study, two models were considered.

Conditional autoregressive (CAR) model:(1)$$Y_{i}=\alpha +\beta X_{i}+u_{i}+v_{i}$$
(2)$$u_{i}|u_{j,\;j\neq i}=N\;\left(\overset{n_{i}}{\underset{j=1}{\sum }}w_{ij}u_{j},\;\frac{\sigma_{u}^{2}}{n_{i}}\right)$$
(3)$$v_{i}\;\sim \;N\;\left(0,\;\sigma_{v}^{2}\right)$$

Specifically, the dependent variable ($Y_{i}$) denoting life expectancy in each spatial unit (*i*) is decomposed into a deterministic part and an unobserved stochastic part (Formula (1)). The deterministic part is explained by the constant (*α*) obeying a normal distribution *α* ~ *N* (0, $\sigma_{\alpha }^{2}$) and a set of covariates $X_{i}$ with associated regression parameters β. For the stochastic part of the model, the spatially structured component $u_{i}$ is normally distributed (Formula (2)), whilst the unstructured component $v_{i}$ is also normally distributed (Formula (3)). $\sum_{j=1}^{n_{i}}w_{ij}u_{j}$ and $\frac{\sigma_{u}^{2}}{n_{i}}$ represent the mean and the variance of the spatial effects of region *i* affected by *j* adjacent regions, respectively. $w_{ij}$ is the inverse distance spatial weight matrix. The parameters and hyperparameters in the model were selected from non-informative prior distributions: log($\sigma_{\alpha }^{2}$)~ logGamma (1, 0.1), log($1/\sigma_{\beta }^{2}$)~ logGamma (1, 0.1), log($1/\sigma_{u}^{2}$)~ logGamma (1, 0.05), and log($1/\sigma_{v}^{2}$)~ logGamma (1, 0.05).

Independent identically distributed (IID) model:(4)$$Y_{i}=\alpha +\beta X_{i}+v_{i}$$

The IID model only considers unstructured random effects ($v_{i}$) (Formula (4)). The prior distributions of the parameters and hyperparameters in the model were chosen in a similar manner to that in the CAR model. The computations were carried out using Integrated Nested Laplace Approximations. The global autocorrelation of spatial effects (Global Moran’s *I* index) was used to determine whether spatially structured random effects needed to be taken into account, which further determined the final model to be used. The model performance was assessed using the adjusted *R*^2^.

To explore the potential association between hypoxia and causes of death, we firstly used a hierarchical cluster analysis (HCA) to identify subgroups with similarity in age-standardized YLLs per 100,000 population for the top 20 level 3 causes. We assumed that hypoxia may lead to higher YLLs of certain causes of death, thus lowering the life expectancy. Therefore, we compared the YLLs of each cause of death and the oxygen concentration between the subgroups. Following this, a univariate analysis of variance (ANOVA) or a nonparametric Kruskal–Wallis test (for the data on age-standardized YLLs per 100,000 population did not obey the prerequisites for ANOVA) was used to test the differences in oxygen concentration and age-standardized YLLs per 100,000 population for each cause of death among the clusters, followed by a pairwise test or pairwise Wilcoxon rank test with Bonferroni correction, respectively.

Bayesian regression analyses were implemented in the R-INLA package [31] within the open-source R software environment [32]. Other statistical analyses were performed using the IBM SPSS statistics 25 software or R (version 3.6.3), and the significance level was 0.05. Maps and figures were drawn using ArcMap 10.2 or R (version 3.6.3).

## 3. Results

Our findings revealed large variances in the geographical distributions of both oxygen concentration and life expectancy in these datasets. The medians (upper and lower quartiles) of oxygen concentration were 155.18 (143.99–158.07) mmHg and 158.34 (152.63–159.21) mmHg at the provincial and municipal levels, respectively. Among the provinces, the coastal and the northeastern areas had higher oxygen concentrations than the western areas (see Figure 1a). The means ± standard deviations of life expectancy at the provincial and municipal levels were (76.48 ± 3.76) years and (79.35 ± 2.62) years, respectively. Higher life expectancy was mostly distributed in northeastern, eastern, and central China, while western provinces, such as Qinghai and Tibet, tended to have lower life expectancy (see Figure 1c). Shanghai had the highest life expectancy while Tibet had the lowest. The municipal-level data also show a similar pattern as the provincial-level data (see Figure 1b,d). Scatter plots show a likely positive association between life expectancy and oxygen concentration (see Figure 1e,f). The summaries of other potentially relevant factors are shown in Appendix A. The life expectancy differed significantly between the group with the higher oxygen concentration compared to the one with lower oxygen concentration at both provincial and municipal (both p<0.05) levels (see Table 1).

The Global Moran’s I indices of spatial effects of life expectancy between the provinces and between the municipalities were 0.472 (p<0.001) and 0.106 (p=0.238), respectively (see Appendix A). Thus, the final models for the provincial-level and municipal-level data were the CAR model and the IID model, respectively. The association between life expectancy and oxygen concentration remained statistically significant after controlling for potential confounding effects using regression analysis (see Table 2). The life expectancy at the provincial level increased by 0.15 (95% CI: 0.10–0.19) years, while at the municipal level, it increased by 0.17 (95% CI: 0.12–0.22) years, with each additional unit (mmHg) of oxygen concentration, suggesting that oxygen concentration may have a positive effect on life expectancy.

According to the dendrogram showing the hierarchical cluster analysis (HCA), we identified three clusters of provinces (see Figure 2). Oxygen concentration and age-standardized YLLs per 100,000 population of several causes in the three clusters demonstrated significant differences (see Table 3, placed at the end of the manuscript due to size). Cluster 3, mainly located in the western part of China with high altitude (see Figure 3), had a much lower level of oxygen concentration (126.11 (105.73–141.17) mmHg), compared to cluster 1 and cluster 2 (with oxygen concentration 154.81 (153.45–158.83) mmHg and 152.69 (147.17–157.00) mmHg, respectively). Several causes of the age-standardized YLLs per 100,000 population (i.e., lower respiratory infection, neonatal disorders, hypertensive heart disease, chronic obstructive pulmonary disease (COPD), cirrhosis and other chronic liver disease (CLD), and chronic kidney disease (CKD)) were significantly higher in cluster 3 than in the other two clusters (adjusted p<0.05).

## 4. Discussion

In this study, we quantified a positive relationship between oxygen concentration and life expectancy and suggested that this relationship may be mediated by diseases potentially related to hypoxia. Our findings provide an initial exploration and evidence for the need for further exploration of the effect of oxygen on aging and life span.

Our results showed that one mmHg higher oxygen concentration was associated with 0.15 (95% BCI: 0.10–0.19) years higher life expectancy at the provincial level and 0.17 (95%BCI: 0.12–0.22) years higher at the municipal level, when controlled for several important potential confounders. Interestingly, the quantitative outcomes, based on our use of both provincial- and municipal-level data, were quite similar. We also adopted multiple linear regression models without spatial random effects, the results of which were quite similar to those of the Bayesian regression models (see Appendix A), suggesting that our findings were robust. The HCA suggested that areas with different levels of oxygen concentration showed different characteristics in terms of causes of death.

In fact, our findings were consistent with the results from several other studies. Burtscher and his colleagues (2013) pointed out that one of the most important prerequisites for anti-aging in humans was aerobic exercise capacity [33],while this capacity was found to be decreased when exposed to hypoxia [34]. Kauppila et al. (2017) demonstrated that moderate regulation of oxygen intake could promote the recovery of vitality in mammals to a certain extent, suggesting that oxygen supply could be a regulator of aging [35,36]. All the above studies showed a positive effect of oxygen in promoting health and alleviating aging. In addition, not surprisingly, our findings suggested that economic levels and medical resources also had a positive impact on life expectancy, and this was consistent with previous studies [37,38]. Some previous studies reported that education was a predictor of life expectancy [39], but in our models, educational situation appeared to have no significant effect on life expectancy. The role of education might be weakened due to the inclusion of GDP per capita in the analysis, as this had a significant correlation.

At present, a large number of people live in high altitude areas (e.g., Tibet and Qinghai) and may, therefore, be exposed to chronic hypoxic conditions, which could affect their life expectancy and overall health, as suggested by our current study and other studies [19,40]. Significant differences in age-standardized YLLs per 100,000 population of several important causes were found between the clusters characterized by different oxygen levels in our study, which suggested that hypoxia may be an influential factor on many important causes of mortality. This has been supported by multiple studies. Martin and Bhattacharya found that hypoxia contributed to the progression and severity of COPD or lower respiratory infection, respectively [30,41]. Similarly, several studies had shown that hypoxia was closely related to the pathogenesis of CKD [42,43], stroke [14,44], cirrhosis, and other CLD [39]. During pregnancy, the mother or the baby is exposed to a hypoxic environment, which may affect the normal development of the child’s brain [45,46], heart [47], or other organs [48], increasing the risk for the development of neonatal diseases [46]. Interestingly, our results showed that the age-standardized YLLs per 100,000 population of tracheal, bronchus, and lung cancers were significantly lower in the cluster of provinces with lower oxygen concentration (see Table 3), which seemed to conflict with the findings that hypoxia may have a general negative impact on the pulmonary system. However, our results were consistent with the findings by Shi and Zhou, which showed that the age-standardized disease burden for lung cancer was significantly lower in Tibet, the province with the lowest oxygen concentration, than that in other provinces [49,50]. These findings were also supported by Ziółkowska-Suchanek, who found that hypoxia-induced FAM13A silencing has a negative effect on lung cancer cell proliferation [51]. Further exploration of these ideas is needed.

To narrow the regional life expectancy gaps and improve equity, special attention should be paid to improve the overall health of people living in high altitude areas. Interventions in oxygen supplies for populations in hypoxic areas might be possible ways to improve health and extend life span, with several studies supporting this notion [52,53]. However, several related studies have not provided sufficient information to support the implementation of measures of management of oxygen supplies for general populations living at high altitudes. Weitzenblum et al. (2002) found that long-term oxygen therapy (LOT) improved the life expectancy of patients with COPD [54], whilst, in contrast, a study by Guo et al. [55] suggested that LOT was not suitable for non-COPD patients. Guo et al. (2015) set the oxygen supply standard for construction personnel in high altitude areas [55] based on the relationship between construction labor intensity and oxygen consumption, but there were no studies available to provide a reference for oxygen supply standards for other occupational groups or general residents living in high altitude areas. It has been demonstrated that hyperbaric oxygen chambers have the capability of increasing arterial oxygen saturation and attenuating chronic high-altitude hypoxia-related sickness [56]. Nonetheless, the inflexibility and potential risks (e.g., middle ear barotrauma, temporary myopia, and pulmonary dyspnea) might limit the widespread application of this technology in high altitudes areas [57]. Further research may focus on how to provide safe and effective oxygenation interventions for populations living at high altitudes and offer a fascinating direction for future studies. Our results provide an initial exploration and reference for follow-up exploration.

Several limitations exist in our study. Firstly, we used an ecological study due to data availability, which was not able to fully take into account individual-level diversity. Nevertheless, we collected as much available data as possible, including both provincial- and municipal-level data, and the results based on the data from the two levels were consistent. Secondly, the outcomes of this study reflect associations instead of cause–effect relationships. Future studies (e.g., biological or laboratory research) will be needed for better insights into the cause–effect relationship between oxygen levels and aging. Thirdly, numerous studies have shown that aging is closely linked to telomere length [58,59,60] and, unfortunately, there are no related data on telomere length available for the high-altitude regions. Future studies about the variability of telomere length among people living in areas with different oxygen levels may be valuable for further understanding the mediation effect of telomere on the association between oxygen levels and aging. Fourthly, oxygen concentration in this study was not obtained by actual measurement due to limited conditions. Instead, we calculated the partial pressure of oxygen based on a simple formula with atmospheric pressure, which was often adopted for oxygen concentration estimates. We also tested the formula by using the actual measurement data of oxygen concentration from Cha’s study [61]. Our estimates on oxygen concentration using the formula were very consistent with their measurements. Fifthly, other factors, e.g., dietary structure [62], air pollution [63,64,65], and marital status [16], which may correlate with life expectancy, were not considered in the modeling analysis due to the lack of data availability. Nevertheless, the most commonly used potential confounders, i.e., economic level, healthcare provision, and meteorological factors, have been included in this study, while taking into account the effects of unconsidered or unknown confounders as the random effects in the model, and, thus, our conclusion is credible. It is well known that physical activity can help preserve health and extend lifespans [66,67], so the large variation in physical activity levels between older adults can present a confounding factor [68] for aging studies. Thus, further study will be needed to quantify the physical activity levels of older participants, which will help differentiate the effects of aging rather than physical inactivity [68].

## 5. Conclusions

A positive correlation was found between oxygen concentration and life expectancy in China, suggesting that oxygen concentration explains a part of the heterogeneity of life expectancy. Differences in YLLs of important causes of death were found between the province-level clusters characterized by different oxygen levels, implying that oxygen concentration may be an important factor on life expectancy when mediated by diseases potentially related to hypoxia. This study provides an epidemiological basis for follow-up investigations on the effect of oxygen concentration on longevity.

## Figures and Tables

**Figure 1 ijerph-20-01125-f001:**
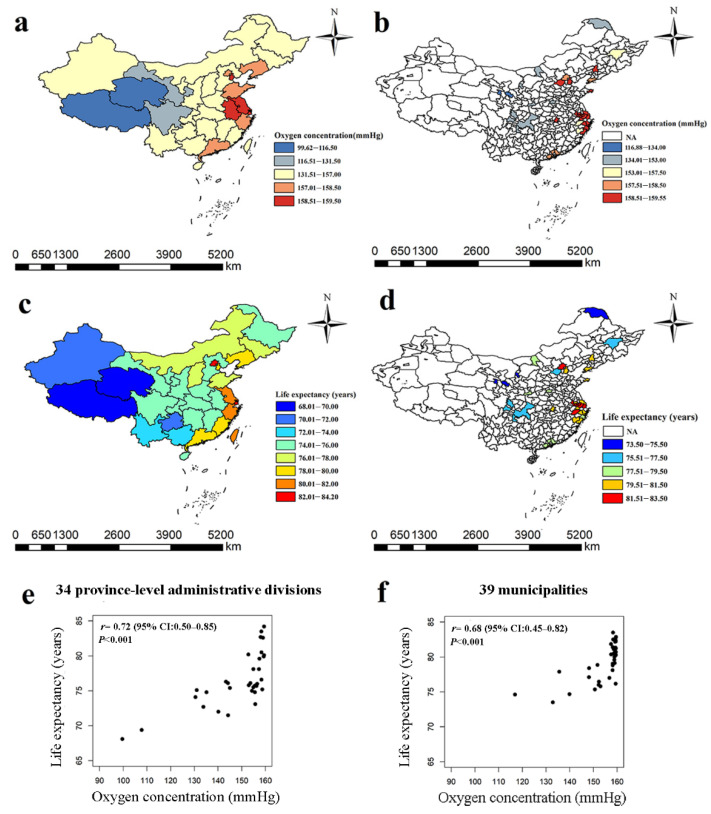
Distribution maps and scatter plots of life expectancy versus oxygen concentration at the provincial and municipal levels in China. (**a**) Oxygen concentration in 34 provinces; (**b**) Oxygen concentration in 39 municipalities; (**c**) Life expectancy in 34 provinces; (**d**) Life expectancy in 39 municipalities; (**e**) Scatter plot of life expectancy and oxygen concentration in 34 provinces; (**f**) Scatter plot of life expectancy and oxygen concentration in 39 municipalities. In (**e**,**f**), r (95% CI) and p value are the results of the Pearson correlation tests calculated by R (version 3.6.3).

**Figure 2 ijerph-20-01125-f002:**
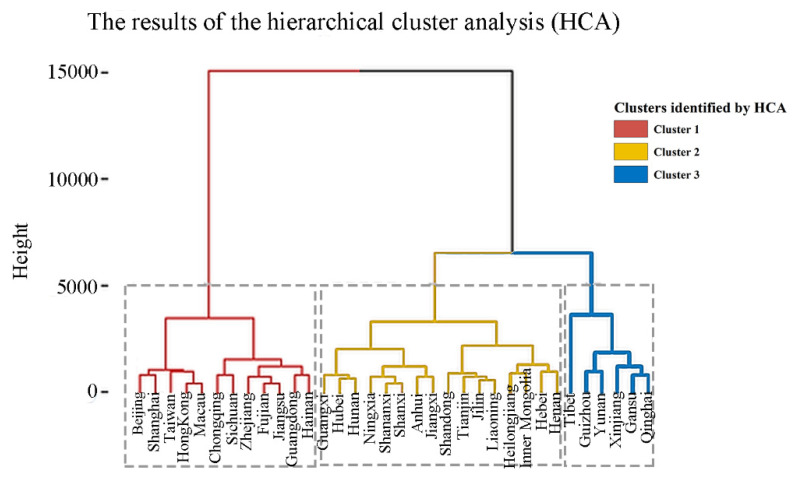
The results of the hierarchical cluster analysis (HCA). Dendrogram of age-standardized years of life lost per 100,000 population data using HCA based on Euclidean distance and the Ward linkage algorithm. Each cluster is identified by a special color. Cluster 1 (*n* = 12): Beijing, Shanghai, Taiwan, Hong Kong, Macau, Chongqing, Sichuan, Zhejiang, Fujian, Jiangsu, Guangdong, and Hainan; Cluster 2 (*n* = 16): Guangxi, Hubei, Hunan, Ningxia, Shananxi, Shanxi, Anhui, Jiangxi, Shandong, Tianjin, Jilin, Liaoning, Heilongjiang, Inner Mongolia, Hebei, and Henan; Cluster 3 (*n* = 6): Tibet, Gansu, Yunnan, Xinjiang, Gansu, and Qinghai.

**Figure 3 ijerph-20-01125-f003:**
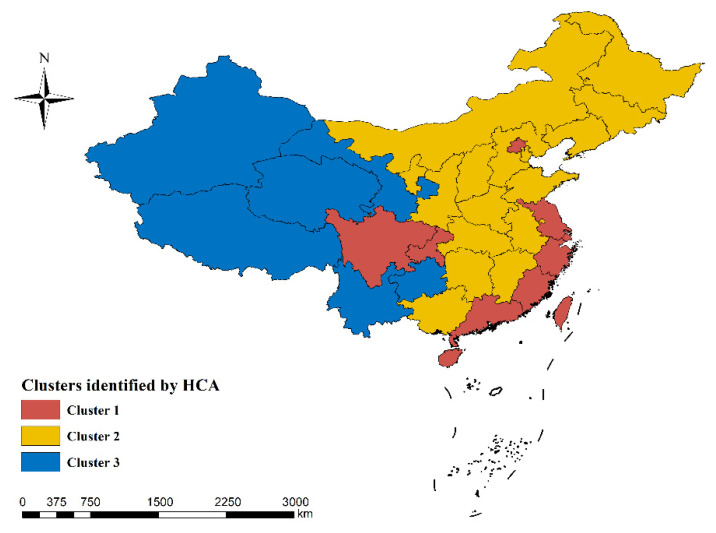
Distribution of the clusters identified by hierarchical cluster analysis (HCA).

**Table 1 ijerph-20-01125-t001:** Comparison of variables between Group A (oxygen concentration ≥ 140 mmHg) and Group B (oxygen concentration < 140 mmHg).

Variables	Group A Median (P_25_–P_75_)	Group B Median (P_25_–P_75_)	Mann–Whitney U	*p* Value
Provincial data	*n* = 28	*n* = 6		
Life expectancy (years)	76.10 (75.43–80.05)	73.4 (69.08–74.88)	13.50	<0.001 *
Oxygen concentration (mmHg)	155.84 (153.14–158.32)	130.76 (105.73–134.17)	0.00	<0.001 *
Health technicians per 1000 people	5.67 (5.29–6.69)	5.22 (4.34–5.87)	52.50	0.159
GDP per capita (10^4^¥)	5.17 (3.96–8.54)	3.44 (2.81–4.19)	27.00	0.008 *
Average years of education (years)	9.32 (8.95–9.53)	8.25 (6.99–8.56)	9.00	0.001 *
Sunshine (h)	6.34 (5.70–7.90)	7.83 (6.29–8.13)	109.00	0.276
Temperature (°C)	18.67 (13.64–22.18)	12.42 (11.19–18.05)	44.00	0.074
Relative humidity (%)	72.96 (66.36–77.93)	54.28 (52.75–73.22)	35.00	0.026 *
Wind speed (m/s)	3.72 (2.81–4.80)	2.81 (2.76–3.29)	48.50	0.110
Municipal data	*n* = 35	*n* = 4		
Life expectancy (years)	80.28 (78.40–81.41)	74.65 (73.78–77.09)	7.00	0.001 *
Oxygen concentration (mmHg)	158.56 (157.43–159.36)	134.24 (120.88–138.82)	140.00	<0.001 *
Health technicians per 1000 people	7.03 (6.22–8.65)	8.69 (7.62–9.32)	106.00	0.102
GDP per capita (10^4^¥)	9.00 (5.90–11.04)	6.30 (5.10–11.80)	57.00	0.576
Average years of education (years)	9.74 (9.08–11.06)	10.10 (9.73–11.40)	53.00	0.460
Sunshine (h)	5.82 (5.08–6.97)	8.32 (8.27–8.40)	128.00	0.004 *
Temperature (°C)	19.20 (15.29–20.24)	11.27 (9.86–12.90)	6.00	0.001 *
Relative humidity (%)	74.36 (71.68–77.09)	56.88 (51.95–60.78)	4.00	<0.001 *
Wind speed (m/s)	2.70 (2.49–3.33)	3.13 (2.17–4.52)	71.00	0.982

* Indicates p<0.05. Abbreviation: (P_25_–P_75_), (lower quartile–upper quartile).

**Table 2 ijerph-20-01125-t002:** Quantitative analysis of the association between life expectancy and oxygen concentration using Bayesian regression model.

	Variables	Mean (β)	95%BCI	Adjusted *R*^2^
Provincial data				0.73
Fixed effect				
*α*	Constant	50.393	(43.018, 57.761)	
$\beta_{1}$	Oxygen concentration (mmHg)	0.145	(0.098, 0.192)	
$\beta_{2}$	GDP per capita (10^4^¥) ^#^	0.877	(0.005, 1.748)	
$\beta_{3}$	Health technicians/1000 people	0.752	(0.246, 1.258)	
Random effect				
$1/\sigma_{u}^{2}$		15.987	(0.547, 66.555)	
$1/\sigma_{v}^{2}$		17.731	(1.085, 64.201)	
Municipal data				0.75
Fixed effect				
*α*	Constant	53.712	(44.871, 62.547)	
$\beta_{1}$	Oxygen concentration (mmHg)	0.170	(0.122, 0.218)	
$\beta_{2}$	GDP per capita (10^4^¥) ^#^	1.415	(0.982, 1.847)	
$\beta_{3}$	Health technicians /1000 people	0.342	(0.053, 0.630)	
$\beta_{4}$	Average years of education (years)	−0.313	(−0.662, 0.036)	
Random effect				
$1/\sigma_{v}^{2}$		0.669	(0.416, 0.979)	

Abbreviation: BCI, Bayesian credible interval. ^#^ GDP per capita are standardized.

**Table 3 ijerph-20-01125-t003:** Comparison of oxygen concentration and age-standardized years of life lost per 100,000 population for several causes of death between the clusters ^λ^.

Oxygen Concentration and Causes of Death	Mean (±SD)/Median (P_25_–P_75_)					Adjusted *p* Value ^Φ^		
	Cluster 1 (*n* = 12)	Cluster 2 (*n* = 16)	Cluster 3 (*n* = 6)	*F*/*H*	*p* Value	Cluster 1 and Cluster 2	Cluster 1 and Cluster 3	Cluster 2 and Cluster 3
Oxygen concentration ^b^	154.81 (153.45–158.83)	152.69 (147.17–157.00)	126.11 (105.73–141.17)	13.107	0.001 *	0.753	0.001 *	0.014 *
Lower respiratory infection ^# b^	292.50 (180.00–420.00)	330.00 (240.00–267.50)	946.67 (625.00–1300.00)	13.817	0.001 *	0.919	0.001 *	0.008 *
Neonatal disorders ^# b^	402.42 (267.50–460.00)	550.63 (342.50–740.00)	1213.33 (860–1600.00)	12.820	0.002 *	0.494	0.001 *	0.026 *
Stomach cancer ^a^	272.5 (±125.41)	394.667 (±122.05)	477.14 (±209.65)	5.819	0.007 *	0.116	0.007 *	0.272
Tracheal, bronchus, and lung cancer ^a^	650.83 (±150.72)	770.67 (±186.57)	458.57 (±165.67)	8.831	0.001 *	0.269	0.036 *	0.001 *
Leukemia ^b^	142.33 (112.50–167.50)	170.63 (160.00–180.00)	195.00 (177.50–207.50)	13.293	0.001 *	0.044 *	0.001 *	0.259
Ischemic heart disease ^a^	820.83 (±232.05)	2026.67 (±435.06)	1800 (±404.15)	36.029	<0.001 *	<0.001 *	<0.001 *	0.360
Stroke ^b^	1067.50 (620.00–1400.00)	2206.25 (1900.00–2300.00)	2800.00 (2300.00–3175.00)	24.865	<0.001 *	<0.001 *	<0.001 *	0.359
Hypertensive heart disease ^# b^	126.83 (83.50–177.50)	258.19 (140.00–347.50)	480.00 (265.00–725.00)	14.808	0.001 *	0.034 *	0.001 *	0.192
COPD ^# b^	570.83 (255.00–637.50)	624.38 (465.00–802.50)	1533.33 (1375.00–1675.00)	14.244	0.001 *	1.000	0.001 *	0.005 *
Cirrhosis and other CLD ^# a^	195.83 (±111.96)	204.38 (±60.33)	475 (±255.24)	11.133	<0.001 *	1.000	<0.001 *	<0.001 *
Falls ^a^	192.58 (±91.86)	175.75 (±78.59)	291.67 (±121.23)	4.106	0.026	0.060	0.067	1.000
CKD ^# b^	158.83 (120.00–180.00)	210.00(160.00–240.00)	320.00 (285.00–352.50)	14.381	0.001 *	0.211	<0.001 *	0.036 *
Congenital ^a^	309.83 (±174.64)	498.75 (±196.80)	588.33 (±311.15)	4.299	0.022 *	0.080	0.040 *	1.000
Road injury ^b^	415.00 (242.50–527.50)	751.88 (592.50–817.50)	951.67 (722.50–1125.00)	22.866	<0.001 *	0.001 *	<0.001 *	0.065

Abbreviations: COPD, chronic obstructive pulmonary disease; CKD, chronic kidney disease; CLD, chronic liver disease; SD, standard deviation; (P_25_–P_75_), (lower quartile–upper quartile). ^λ^ Only the significant results of the ANOVA or the Kruskal–Wallis tests are listed. ^a^ indicates that the data obey the prerequisites for ANOVA, and ^b^ indicates that they do not obey. * indicates *P* < 0.05. ^#^ indicates that the age-standardized years of life lost per 100,000 population of causes of death of cluster 3 are significantly higher than other clusters. ^Φ^ The significance values have been adjusted using the Bonferroni correction for multiple tests.

## Data Availability

All data files are publicly available and the sources can be found in the Appendix A.

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
