# Peer review of "Exploring the Association between Oxygen Concentration and Life Expectancy in China: A Quantitative Analysis"

_ijerph, 2023, doi:10.3390/ijerph20021125_

Round 1
Reviewer 1 Report
The study is well planned, well performer and well described. Minor revision required:
- please provide graphics summarizing the observed results
- please provide recent literature in discussion section
- please describe limitations of the study
Reviewer 2 Report
I appreciate the opportunity of reviewing the manuscript entitled "Exploring the Association between Oxygen Concentration and Life Expectancy in China: A Quantitative Analysis"
Here you can see my comments and recommendations:
Abstract
Line 18: I consider that "a" in "that a one" is not addecuate.
Why do you abbreviate "oxygen concentration" and "years of life lost"
Introduction: You begin talking about molecular processes associated with ageing I expecter to see results focused on that, but I didn't see any one.
Line 46: What does "so on" means?
Line 53: Is "et al's" correct?
Materials and Methods:
You must respect abbreviations along the document
I don't understand section 2.1.3.
Section 2.2.2. Even you present the links for the GDP sources in S1, you should put the principal information of methods here.
Line 115: Cite a reference for the affirmation of that "dropout rates in Chine are low"
Results:
I consider that you made so much statistical analysis of the data, and before being explanatory, it gets to the point of confusing."I couldn't understand Table 3, that it is what I was waiting to read. You focus so much in PO2 association with life expentancy, but I want to understand more why this es more important than the deads by a chronic disease or cancer, for example.
Discussion:
Why do you consider that increase life expentancy by two months each mm Hg is clinically important?
You talk about anti ageing prerequisite in humans was aerobic exercise capacity, and effectively, it is a fact well known, but you never mention something about physical activity levels in your data, that probably are associated with the PO2 reported in the data analysed. Physical funtion of the elderly must have been taken in account too.
I can't understand the new scientific information importance of your manuscript.
Reviewer 3 Report
An interesting study with a potential impact on policies that may be adopted in the future.
In terms of structure, it seems detailed to me. The authors tried to clearly describe the variables in studies, the potential confounders' economic level, healthcare provision and meteorological factors.
Data analysis was detailed and considered these factors in its investigation to make the statistics obtained more robust.
The main criticism is that other factors can influence the results in addition to those that were taken into account. Still, the authors mentioned this in their limitations and used the main variables to be considered.
Discussion carried out considering the state of the existing evidence, although it is scarce, highlights the study's importance.
Round 2
Reviewer 2 Report
I appreciate you for addressing my comments.
I recommend that the authors calmly review the punctuations and syntax thoroughly because there was a lack of quality in that sense, you can even see a paragraph in the discussion without justification in the text and in bold letters.
In addition, English language has to be reviewed by a native speaker, because it continues presenting errors.
